# Causes of Death among Patients with Hepatocellular Carcinoma According to Chronic Liver Disease Etiology

**DOI:** 10.3390/cancers15061687

**Published:** 2023-03-09

**Authors:** Yi-Hao Yen, Kwong-Ming Kee, Wei-Feng Li, Yueh-Wei Liu, Chih-Chi Wang, Tsung-Hui Hu, Ming-Chao Tsai, Yuan-Hung Kuo, Chih-Yun Lin

**Affiliations:** 1Division of Hepatogastroenterology, Department of Internal Medicine, Kaohsiung Chang Gung Memorial Hospital, Kaohsiung 83301, Taiwan; 2Liver Transplantation Center, Department of Surgery, Kaohsiung Chang Gung Memorial Hospital, Kaohsiung 83301, Taiwan; 3Biostatistics Center, Kaohsiung Chang Gung Memorial Hospital, Kaohsiung 83301, Taiwan

**Keywords:** liver cancer, cause-specific mortality, chronic liver disease (CLD)

## Abstract

**Simple Summary:**

Hepatocellular carcinoma (HCC) is a highly aggressive and lethal form of liver cancer, and most patients with HCC die due to HCC-related causes. Although most patients die of HCC-related causes, non–HCC-related death represents a competing event among patients who engage in alcohol use and receive curative treatment and among patients 75 years and older in the hepatitis B virus and all-negative groups who receive curative treatments. All negative was defined as negative for hepatitis C virus, hepatitis B virus, and alcohol-related causes. The results of the current study underscore the importance of assessing and managing underlying comorbidities, especially among certain subgroups of patients with HCC.

**Abstract:**

This study was conducted to determine whether the causes of death among patients with hepatocellular carcinoma (HCC) differ according to chronic liver disease (CLD) etiology. Between 2011 and 2020, 3977 patients who were newly diagnosed with HCC at our institution were enrolled in this study. We determined whether the cause of death was HCC-related and non-HCC-related. For patients with multiple CLD etiologies, etiology was classified using the following hierarchy: hepatitis C virus (HCV) > hepatitis B virus (HBV) > alcohol-related causes > all negative. All negative was defined as negative for HCV, HBV, and alcohol-related causes. Among 3977 patients, 1415 patients were classified as HCV-related, 1691 patients were HBV-related, 145 patients were alcohol-related, and 725 patients were all negative. HCC-related mortality was the leading cause of death, irrespective of etiology. Among patients who underwent curative treatment, HCC-related mortality was the leading cause of death for patients in the HCV, HBV, and all-negative groups, but not for patients in the alcohol-related group. Among patients 75 years and older who underwent curative treatment, HCC-related mortality was the leading cause of death in the HCV but not HBV or all-negative groups. In conclusion, although most patients with HCC die due to HCC-related causes, non-HCC-related mortality represents a competing event in certain patient subgroups. The current study results underscore the importance of assessing and managing underlying comorbidities, particularly among patients with HCC at risk of non-HCC-related mortality.

## 1. Introduction

Hepatocellular carcinoma (HCC) is a common cause of cancer-related death worldwide [1]. Non-alcoholic fatty liver disease (NAFLD) is the fastest growing HCC etiology globally due to the obesity epidemic [2,3]. Although cardiovascular disease is the leading cause of death among patients with NAFLD without HCC [4], whether this remains true among patients with NAFLD-related HCC remains unclear. A retrospective study including a cohort of patients with NAFLD-related HCC who were diagnosed at Veterans Administration (VA) facilities found that 40% of all deaths occurring 3–5 years after treatment were due to non-HCC-related causes among patients with NAFLD-related HCC who received curative treatment. All-cause mortality was driven by non-HCC-related mortality in patients 75 years and older who underwent curative treatments, as the cumulative number of non-HCC-related deaths was higher than HCC-related deaths during the treatment follow-up period. This study enrolled patients treated by the VA, who were mostly men, limiting the generalizability of study findings to women with NAFLD-related HCC [5]. Furthermore, the proportion of non-HCC-related deaths that occur among patients with NAFLD-related HCC compared with other chronic liver disease (CLD) etiologies remains unclear. We aim to clarify this issue.

## 2. Materials and Methods

The Institutional Review Board of Kaohsiung Chang Gung Memorial Hospital, Taiwan, approved this study (reference number: 202201189B0) and waived the need for informed consent due to the retrospective and observational nature of the study design. Data were extracted from Kaohsiung Chang Gung Memorial Hospital’s HCC registry database of prospectively collected and annually updated data.

From 2011 to 2020, 3977 patients who were newly diagnosed with HCC at the institution were enrolled in this study.

### 2.1. Variables of Interest

Patient clinical data, including tumor number and size, imaging-diagnosed tumor–node–metastasis (TNM) stage (based on the 7th edition of the American Joint Committee on Cancer [AJCC]) [6], Barcelona Clinic Liver Cancer (BCLC) stage [7], serum alpha-fetoprotein (AFP) level, presence of liver cirrhosis, Child–Pugh class [8], international normalized ratio (INR), creatinine level, bilirubin level, presence of hepatitis B surface antigen (HBsAg), presence of anti-HCV antibody, alcohol intake (assessed by asking patients how often they have a drink containing alcohol), and HCC diagnostic method (i.e., clinical vs. pathological diagnosis), were prospectively collected from the HCC registry data. Patients with HCV infection were defined based on anti-HCV antibody positivity. Patients with HBV infection were defined based on HBsAg positivity. Patients were defined as consuming alcohol if they reported regularly partaking in alcoholic beverages. Patients were classified as all negative if they were negative for HCV, HBV, and alcohol intake. For patients with multiple etiologies, classification was performed using the following hierarchy: HCV > HBV > alcohol-related causes > all negative [9]. Demographic information included height, weight, age, and sex. Tumor number (solitary vs. multiple) was determined based on imaging results. Tumor size was determined according to pathological examinations in patients who underwent surgery or imaging findings in patients who underwent non-surgical treatments. The presence of cirrhosis was indicated by an Ishak score [10] of 5 or 6 in patients who underwent surgery or imaging results in patients who underwent non-surgical treatments. Cirrhosis was indicated if imaging results showed small liver size, nodular liver surface, and the presence of regeneration nodules [11]. BCLC stages were defined according to the original definitions, and BCLC stage A was defined using the Milan criteria [12]. The cause of death was derived from death certificates. Curative treatment was defined as liver transplantation, liver resection, or ablation. Non-curative treatments included transcatheter arterial embolization (TAE)/transcatheter arterial chemoembolization (TACE), target therapy (i.e., sorafenib or lenvatinib), systemic chemotherapy, radiation therapy, and best supportive care.

Raw data for the cohort involved in this study is available via the following digital object identifier: https://www.dropbox.com/scl/fi/9y5d6re7n4cjt6viqakx4/raw-data-for-submission-n-3977.xlsx?dl=0&rlkey=szktna5d1pmnho47afrfrmoti (accessed on 6 March 2023).

### 2.2. Statistical Analysis

Variables are presented as the number and percentage or the median and interquartile range. The Chi-square test was used to compare categorical variables. The Kruskal–Wallis test was used to compare continuous variables. We calculated cumulative and cause-specific mortality within 5 years after HCC diagnosis. The cumulative HCC-related mortality analysis considered non-HCC-related and unknown causes of death as competing risks. All statistical analyses were performed using SPSS version 25.0 and SigmaPlot 14.0. Two-tailed significance values were applied, with significance defined as *p* < 0.05.

## 3. Results

### 3.1. Characteristics of Patients According to CLD Etiology

The all-negative group was older (*p* < 0.001) than the other groups. The proportion of men was smaller in the HCV group (*p* < 0.001) than in the other groups. The proportion of HCC diagnosed pathologically was smaller in the alcohol-related group (*p* < 0.001) than in the other groups. The tumor size was larger in the all-negative group (*p* < 0.001) than in the other groups. The proportions of TNM stages I and II were larger in the HCV group (*p* < 0.001) than in the other groups. The proportions of BCLC stages 0 and A were larger in the HCV group (*p* < 0.001) than in the other groups. The body mass index (BMI) was higher in the all-negative group (*p* = 0.048) than in the other groups. The proportion of AFP ≥20 ng/dl was larger in the HBV group (*p* = 0.001) than in the other groups. The proportion without cirrhosis was larger in the all-negative group (*p* < 0.001) than in the other groups. The creatinine level was higher in the all-negative group (*p* < 0.001) than in the other groups. The bilirubin level was higher in the alcohol-related group (*p* < 0.001) than in the other groups. The INR level was higher in the alcohol-related group (*p* < 0.001) than in the other groups. The proportion of Child–Pugh class A was smaller in the alcohol-related group (*p* < 0.001) than in the other groups. The proportion of patients who underwent resection was larger in the HBV group than in the other groups. The proportion of patients who underwent ablation was larger in the HCV group than in the other groups. The proportion of patients who underwent TAE/TACE was larger in the alcohol-related group than in the other groups (*p* < 0.001). However, no significant difference in tumor number was observed between the groups (Table 1).

### 3.2. Cause-Specific Mortality According to CLD Etiology

Figure 1 shows the cumulative mortality risk according to CLD etiology for all patients. HCC-related mortality was the leading cause of death among all patients, irrespective of CLD etiology (Figure 1A–D). The proportion of HCC-related death was largest among patients with HBV-related HCC and smallest among patients in the alcohol-related and all-negative groups. Among patients with HBV-related HCC, HCC contributed to 73.5% (424 of 577) of all deaths within 3 years. Approximately 12.5% of deaths within 3 years were attributed to non-HCC-related causes, and 14.0% were attributed to unknown causes. We found similar mortality outcomes among patients with HBV-related HCC at 1, 3, and 5 years. In the all-negative group, HCC contributed to 62.2% (186 of 299) of deaths within 3 years, non-HCC-related mortality contributed to approximately 25.8% of deaths within 3 years, and 12.0% was attributed to unknown causes. We found similar mortality outcomes among all-negative patients at 1, 3, and 5 years (Table 2).

### 3.3. Cause-Specific Mortality among Patients Who Received Curative Treatments According to CLD Etiology

Figure 2 shows the cumulative mortality risk according to CLD etiology among patients who underwent curative treatments. HCC-related mortality was the leading cause of death in the HCV (Figure 2A), HBV (Figure 2B), and all negative (Figure 2D) groups. However, the cumulative non-HCC-related mortality rate was higher than the HCC-related mortality rate for deaths that occurred within 5 years of treatment in the alcohol-related group (Figure 2C). The non-HCC-related mortality rates were higher than the HCC-related mortality rates at 1, 3, and 5 years in the alcohol-related group. By contrast, in the remaining three groups, the HCC-related mortality rates were higher than the non-HCC-related mortality rates at 1, 3, and 5 years (Table 3).

### 3.4. Cause-Specific Mortality in Patients Who Received Curative Treatments According to CLD Etiology and Stratified by Age

Figure 3 shows the cumulative mortality risk according to CLD etiology in patients younger than 75 years who underwent curative treatments. HCC-related mortality was the leading cause of death for HCV (Figure 3A), HBV (Figure 3B), and all-negative groups (Figure 3D). By contrast, HCC-related mortality and non-HCC-related mortality were similar in the alcohol-related group (Figure 3C).

Figure 4 shows the cumulative mortality risk according to CLD etiology in patients 75 years and older who underwent curative treatments. HCC-related mortality was the leading cause of death for the HCV group (Figure 4A). However, HCC-related mortality and non-HCC-related mortality rates were similar in the HBV (Figure 4B) and all-negative groups (Figure 4C). Only three patients were included in the alcohol-related group, preventing the analysis of this group.

### 3.5. Cause-Specific Mortality in Patients Who Received Non-Curative Treatments According to CLD Etiology

HCC-related mortality was the leading cause of death among patients who underwent non-curative treatments, irrespective of CLD etiology (Figure 5A–D).

## 4. Discussion

In the current study, we found that although most patients with HCC die due to HCC-related causes, non-HCC-related mortality represents a competing event for certain subgroups of patients with HCC, including patients who partake in regular alcohol use who receive curative treatment and HBV and all-negative patients 75 years and older who receive curative treatment.

Tumor stage, the Eastern Cooperative Oncology Group performance status, treatment strategy, and liver disease severity are associated with mortality in patients with HCC according to the BCLC guidelines [7]. A recent study enrolled 10,826 patients with HCC from the Surveillance, Epidemiology, and End Results-Medicare database and found that the receipt of curative treatment was the strongest predictor of survival beyond 5 years among HCC patients [13]. In the current study, we analyzed the cause of death in patients stratified according to the receipt of curative treatment. As expected, HCC-related mortality was the leading cause of death for all patients and in patients who received non-curative treatments, irrespective of CLD etiology. However, among patients who received curative treatments, HCC-related mortality was the leading cause of death in the HCV, HBV, and all-negative groups but not in the alcohol-related group. Alcohol use impacts the outcomes of several diseases and injuries. The largest numbers of deaths attributable to heavy alcohol intake are associated with cardiovascular disease, followed by injuries, cirrhosis, and cancer [14]; the higher risks of these etiologies among alcohol users may explain the higher rate of non-HCC-related mortality observed for patients in the alcohol-related group who underwent curative treatment.

Aging is a prognostic factor for poor outcomes in most chronic diseases, including HCC [15]. The association between aging and poor prognosis in HCC patients could be due to older patients having higher risks of severe comorbidities. The World Health Organization has stated that people 65 years old and older can be defined as elderly in developed countries (https://www.who.int/healthinfo/survey/ageingdefnolder/en, accessed on 6 March 2023). Moreover, the definitions of “elderly” are changing, especially as life expectancies in many developed countries, including Taiwan, now exceed 80 years. Most people in their 60s and early 70s remain active, whereas people usually become frail after passing 75 years of age [16].

In this study, we defined elderly patients as those 75 years or older. Older patients are associated with higher risks of severe comorbidities, which could explain the similar rates of HCC-related and non-HCC-related death among patients 75 years and older who underwent curative treatments in the all-negative and HBV groups. However, the leading cause of death in the HCV group remained HCC. The proportion of cirrhotic patients was larger in the HCV group than in the other groups, and the risk of recurrent HCC has been associated with cirrhosis, suggesting that the risk of recurrent HCC may also be higher in the HCV group than in the other groups, as cirrhosis is a well-known risk factor for HCC recurrence after curative treatment [17].

The results of the current study underscore the importance of assessing and managing underlying comorbidities in patients with HCC, especially among patients who engage in alcohol use who receive curative treatments and all-negative and HBV-positive patients 75 years and older who receive curative treatments.

Compared to groups with other CLD etiologies, the all-negative group was the oldest and presented with the largest tumor size, the smallest proportion of early-stage HCC (i.e., BCLC stages 0 and A), the highest creatinine level, the highest BMI, and the largest proportion of non-cirrhotic livers, which is consistent with the characteristics of NAFLD-related HCC [9,18,19]. Patients with NAFLD-related HCC presented with larger tumors at later disease stages than patients with virus-related HCC [19], which may be due to a low HCC surveillance rate among NAFLD patients. A previous study reported that NAFLD is the leading cause of non-cirrhotic HCC [9]. NAFLD is a multisystem disease affecting extra-hepatic organs and increasing the risks of developing cardiovascular and chronic kidney disease [20], which could explain the characteristics of NAFLD-related HCC patients. The etiologies of non-viral HCC are most commonly alcohol use and NAFLD, although rarer etiologies have been identified, such as primary biliary cirrhosis, primary sclerosing cholangitis, and autoimmune hepatitis [9]. Therefore, we expected the majority of patients in the all-negative group would have NAFLD-related HCC. A recent study reported that among NAFLD-related HCC patients, most mortality (72.2% at 3 years) was attributable to HCC, although, among NAFLD-related HCC patients who underwent curative treatment, non-HCC-related mortality accounted for 40% of all deaths between 3 and 5 years after treatment [5]. In the current study, 62.2% of all death in the all-negative group were attributable to HCC at 3 years, and non-HCC-related mortality accounted for 37.5%–38.5% of all deaths between 3 and 5 years following curative treatment in the all-negative group, which is comparable to the previous study [5].

A strength of the present study was the use of a large cohort of patients with HCC associated with prospectively collected data and limited missing data. However, the study also has several limitations. First, the study lacked a complete list of comorbidities, which are the leading causes of non-HCC-related mortality. Multiple comorbidity indices (e.g., the cirrhosis-related comorbidity score, the National Cancer Institute Comorbidity Index, etc.) have been shown to have value for predicting mortality among HCC patients [5,19]. In the current study, we used older age (75 years or older) as a simple and objective surrogate for severe comorbidities. Furthermore, we did not have data on hepatic steatosis and metabolic risk factors (e.g., hypertension, dyslipidemia, central obesity, and hyperglycemia) [4]. Therefore, we could not define NAFLD in the present study. Third, this study was conducted as a retrospective and monocentric study. Fourth, the case number in the alcohol-related group was limited. Fifth, the etiology of liver disease is not properly defined. We defined the viral etiology just on the basis of anti-HCV antibodies or of HBsAg positivity, without giving information about antiviral therapies that might have caused suppression of HBV replication or HCV clearance. Finally, causes of death were derived from death certificates, which are not enough accurate for the study purposes.

## 5. Conclusions

Although most patients die of HCC-related causes, non-HCC-related death represents a competing event among patients who engage in alcohol use and receive curative treatment and among patients 75 years and older in the HBV and all-negative groups who receive curative treatments. The results of the current study underscore the importance of assessing and managing underlying comorbidities, especially among certain subgroups of patients with HCC.

## Figures and Tables

**Figure 1 cancers-15-01687-f001:**
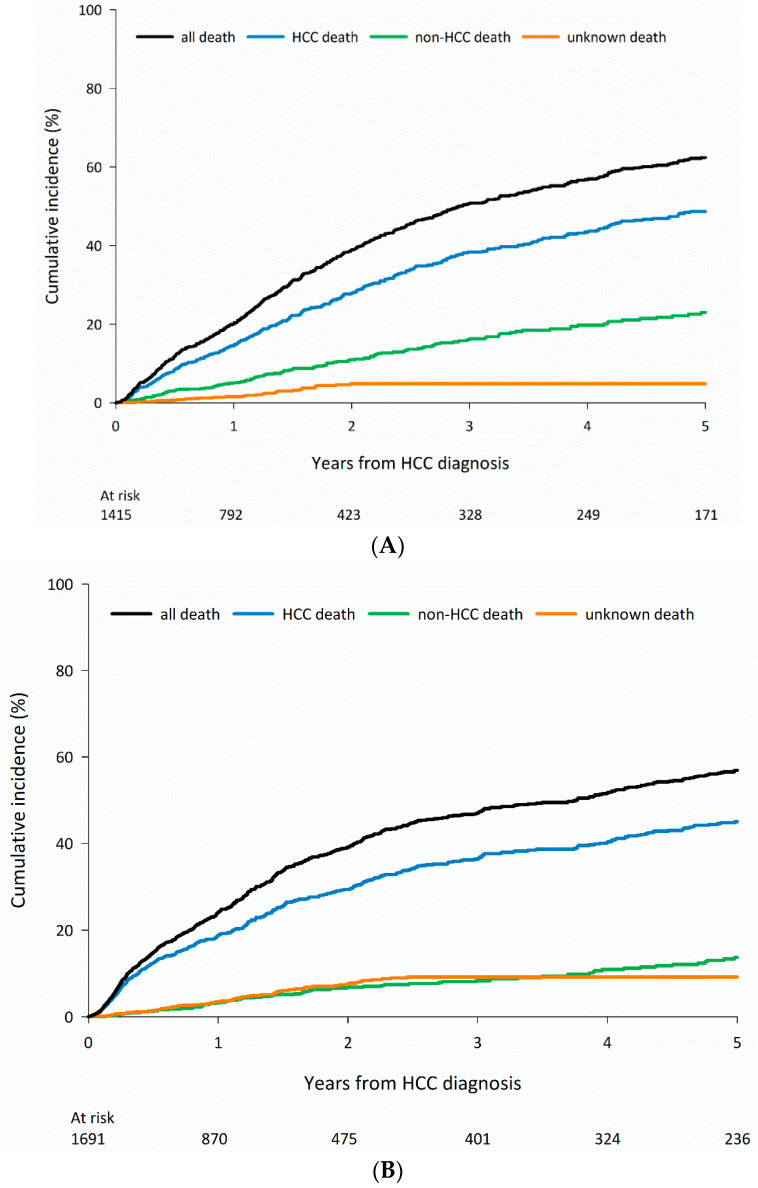
Cumulative mortality risk for hepatocellular carcinoma according to chronic liver disease etiology: (**A**) hepatitis C virus; (**B**) hepatitis B virus; (**C**) alcohol use; and (**D**) all negative (defined as no evidence of virus infection or alcohol use).

**Figure 2 cancers-15-01687-f002:**
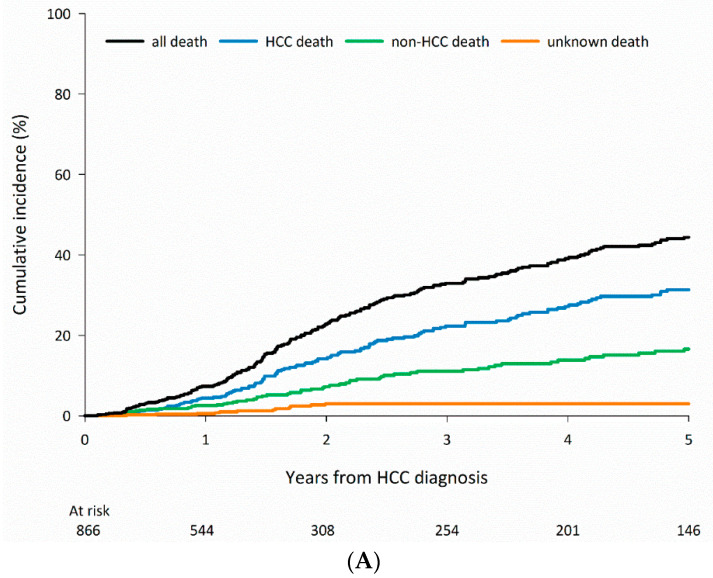
Cumulative mortality risk for hepatocellular carcinoma among patients who received curative treatments according to chronic liver disease etiology: (**A**) hepatitis C virus; (**B**) hepatitis B virus; (**C**) alcohol use; and (**D**) all negative (defined as no evidence of virus infection or alcohol use).

**Figure 3 cancers-15-01687-f003:**
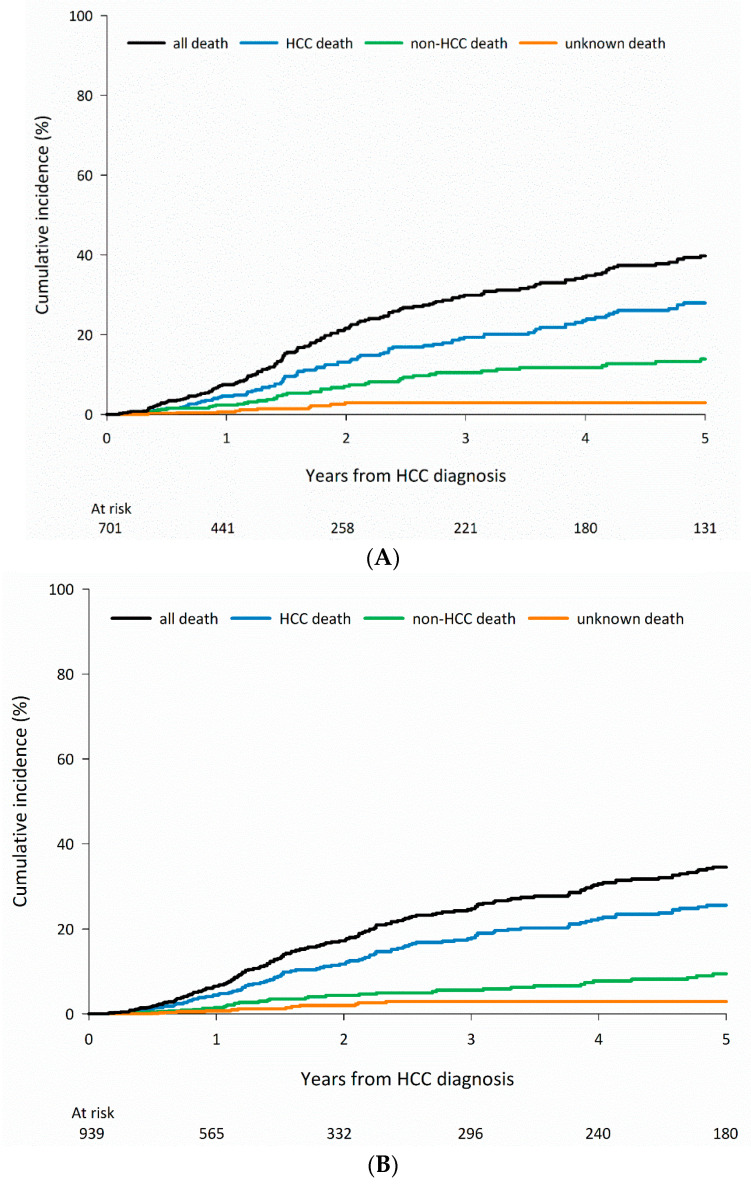
Cumulative mortality risk for hepatocellular carcinoma among patients younger than 75 years who received curative treatments according to chronic liver disease etiology: (**A**) hepatitis C virus; (**B**) hepatitis B virus; (**C**) alcohol use; and (**D**) all negative (defined as no evidence of virus infection or alcohol use).

**Figure 4 cancers-15-01687-f004:**
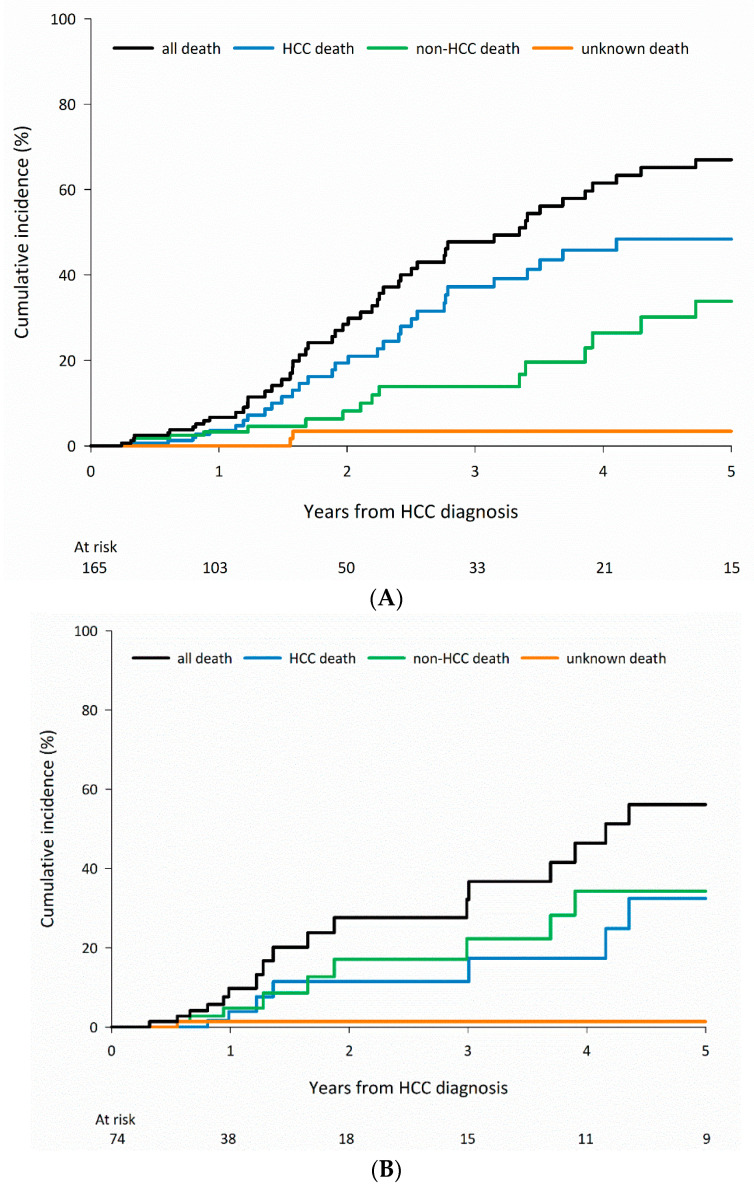
Cumulative mortality risk for hepatocellular carcinoma among patients 75 years and older who received curative treatments according to chronic liver disease etiology: (**A**) hepatitis C virus; (**B**) hepatitis B virus; and (**C**) all negative (defined as no evidence of virus infection or alcohol use).

**Figure 5 cancers-15-01687-f005:**
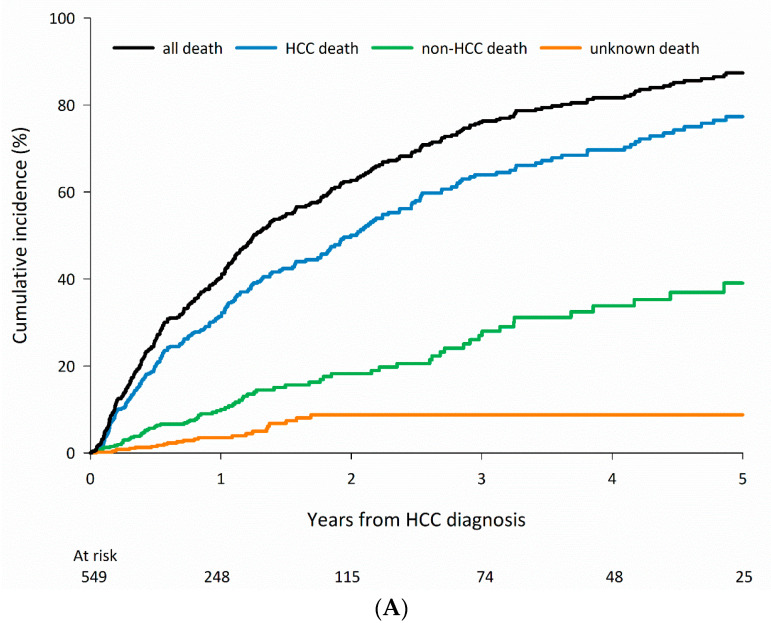
Cumulative mortality risk for hepatocellular carcinoma among patients who received non-curative treatments according to chronic liver disease etiology: (**A**) hepatitis C virus; (**B**) hepatitis B virus; (**C**) alcohol use; and (**D**) all negative (defined as no evidence of virus infection or alcohol use).

**Table 1 cancers-15-01687-t001:** Characteristics of patients according to etiologies of chronic liver disease.

	HCV, *n* = 1415	HBV, *n* = 1691	Alcohol, *n* = 145	All Negative, *n* = 725	*p*
Age (years)	66 (60–73)	60 (52–67)	59 (51.5–65)	68 (60–76)	<0.001
Male	833 (58.9%)	1394 (82.4%)	141 (97.2%)	503 (69.4%)	<0.001
Method of HCC diagnosis					<0.001
Clinical	559 (39.5%)	631 (37.3%)	68 (46.9%)	231 (31.9%)	
Pathological	856 (60.5%)	1061 (62.7%)	77 (53.1%)	494 (68.1%)	
Tumor size (mm)	30 (21–50)	35 (23–75)	32 (21.5–82.5)	48 (28–95)	<0.001
7th edition AJCC stage					<0.001
1	742 (52.4%)	826 (48.8%)	60 (41.4%)	328 (45.2%)	
2	317 (22.4%)	283 (16.7%)	39 (26.9%)	115 (15.9%)	
3	250 (17.7%)	391 (23.1%)	26 (17.9%)	183 (25.2%)	
4	89 (6.3%)	164 (9.7%)	17 (11.7%)	81 (11.2%)	
Unknown	17 (1.2%)	28 (1.7%)	3 (2.1%)	18 (2.5%)	
Tumor number by imaging studies					0.436
Single	870 (61.5%)	1032 (61.0%)	79 (54.5%)	443 (61.1%)	
Multiple	545 (38.5%)	660 (39.0%)	66 (45.5%)	282 (38.9%)	
BCLC stage					<0.001
0	214 (15.1%)	220 (13.0%)	18 (12.4%)	47 (6.5%)	
A	581 (41.1%)	592 (35.0%)	45 (31.0%)	205 (28.3%)	
B	240 (17.0%)	344 (20.3%)	34 (23.4%)	190 (26.2%)	
C	283 (20.0%)	421 (24.9)	36 (24.8)	215 (29.7%)	
D	69 (4.9%)	82 (4.8%)	9 (6.2%)	45 (6.2%)	
Unknown	28 (2.0%)	33 (2.0%)	3 (2.1%)	23 (3.2%)	
BMI (kg/m^2^)	24.5 (22.3–27.3)	24.5 (22.1–27.3)	24.8 (22.0–27.4)	25.0 (22.6–28.0)	0.048
AFP					0.001
≥20 ng/ml	741 (52.4%)	902 (53.3%)	64 (44.1%)	329 (45.4%)	
<20 ng/ml	674 (47.6%)	790 (46.7%)	81 (55.9%)	396 (54.6%)	
Cirrhosis					<0.001
Yes	1062 (75.3%)	1157 (68.4%)	100 (69.4%)	444 (61.8%)	
No	348 (24.7%)	534 (31.6%)	44 (30.6%)	275 (38.2%)	
Unknown					
Creatinine (mg/dL)	1.0 (0.8–1.3)	1.0 (0.8–1.2)	1.1 (0.9–1.4)	1.1 (0.8–1.5)	<0.001
Total bilirubin (mg/dL)	1.1 (0.8–1.6)	1.0 (0.8–1.6)	1.3 (0.8–2.3)	1.0 (0.7–1.5)	<0.001
INR	1.0 (1.0–1.1)	1.0 (1.0–1.1)	1.0 (1.1–1.2)	1.0 (1.0–1.1)	<0.001
Child Pugh class					<0.001
A	1125 (79.5%)	1398 (82.6%)	100 (69.0%)	591 (81.5%)	
B	229 (16.2%)	212 (12.5%)	39 (26.9%)	100 (13.8%)	
C	38 (2.7%)	62 (3.7%)	6 (4.1%)	15 (2.1%)	
Unknown	23 (1.6%)	20 (1.2%)	0	19 (2.6%)	
Treatment					<0.001
Transplant	54 (3.8%)	58 (3.4%)	5 (3.4%)	16 (2.2%)	
Resection	398 (28.1%)	640 (37.8%)	38 (26.2%)	243 (33.5%)	
Ablation	414 (29.3%)	316 (18.7%)	33 (22.8%)	133 (18.3%)	
Best supportive care	73 (5.2%)	87 (5.1%)	12 (8.3%)	43 (5.9%)	
Chemotherapy	10 (0.7%)	35 (2.1%)	2 (1.4%)	14 (1.9%)	
TAE/TACE	328 (23.2%)	318 (18.8%)	39 (26.9%)	168 (23.2%)	
Target therapy	95 (6.7%)	186 (11.0%)	11 (7.6%)	77 (10.6%)	
Radiation therapy	43 (3.0%)	52 (3.1%)	5 (3.4%)	31 (4.3%)	

AFP, alpha-fetoprotein; BMI, body mass index; HCV, hepatitis C virus; HBV, hepatitis B virus; INR, international normalized ratio; AJCC, American Joint Committee on Cancer; BCLC, Barcelona clinic liver cancer; TACE, Transcatheter Arterial Chemoembolization; TAE, Transcatheter Arterial embolization.

**Table 2 cancers-15-01687-t002:** Cause-specific mortality among all patients.

Group	Cause of Mortality	1-Year	3-Year	5-Year
Total, *N* = 3976	Any cause	883 (22.2)	1426 (35.9)	1593 (40.1)
	HCC-related	639 (72.4)	987 (69.2)	1098 (68.9)
	Non-HCC-related	158 (17.9)	283 (19.8)	339 (21.3)
	Unknown	86 (9.7)	156 (10.9)	156 (9.8)
HCV, *n* = 1415	Any cause	269 (19.0)	493 (34.8)	561 (39.6)
	HCC-related	189 (70.3)	339 (68.8)	385 (68.6)
	Non-HCC-related	62 (23.0)	118 (23.9)	140 (25.0)
	Unknown	18 (6.7)	36 (7.3)	36 (6.4)
HBV, *n* = 1691	Any cause	384 (22.7)	577 (34.1)	644 (38.1)
	HCC-related	295 (76.8)	424 (73.5)	472 (73.3)
	Non-HCC-related	42 (10.9)	72 (12.5)	91 (14.1)
	Unknown	47 (12.2)	81 (14.0)	81 (12.6)
Alcohol, *n* = 145	Any cause	34 (23.4)	57 (39.3)	66 (45.5)
	HCC-related	23 (67.6)	38 (66.7)	41 (62.1)
	Non-HCC-related	10 (29.4)	16 (28.1)	22 (33.3)
	Unknown	1 (2.9)	3 (5.3)	3 (4.5)
All negative, *n* = 725	Any cause	196 (27.0)	299 (41.2)	322 (44.4)
	HCC-related	132 (67.3)	186 (62.2)	200 (62.1)
	Non-HCC-related	44 (22.4)	77 (25.8)	86 (26.7)
	Unknown	20 (10.2)	36 (12.0)	36 (11.2)

HCC, hepatocellular carcinoma; HCV, hepatitis C virus; HBV, hepatitis B virus.

**Table 3 cancers-15-01687-t003:** Cause-specific mortality among patients who underwent curative treatment.

Group	Cause of Mortality	1-Year	3-Year	5-Year
Total, *N* = 2347	Any cause	154 (6.6)	402 (17.1)	496 (21.1)
	HCC-related	88 (57.1)	242 (60.2)	302 (60.9)
	Non-HCC-related	51 (33.1)	122 (30.3)	156 (31.5)
	Unknown	15 (9.7)	38 (9.5)	38 (7.7)
HCV, *n* = 866	Any cause	58 (6.7)	164 (18.9)	202 (23.3)
	HCC-related	34 (58.6)	101 (61.6)	126 (62.4)
	Non-HCC-related	20 (34.5)	50 (30.5)	63 (31.2)
	Unknown	4 (6.9)	13 (7.9)	13 (6.4)
HBV, *n* = 1013	Any cause	61 (6.0)	144 (14.2)	184 (18.2)
	HCC-related	39 (63.9)	94 (65.3)	122 (66.3)
	Non-HCC-related	15 (24.6)	35 (24.3)	47 (25.5)
	Unknown	7 (11.5)	15 (10.4)	15 (8.2)
Alcohol, *n* = 76	Any cause	5 (6.6)	14 (18.4)	19 (25.0)
	HCC-related	0 (0.0)	6 (42.9)	7 (36.8)
	Non-HCC-related	4 (80.0)	7 (50.0)	11 (57.9)
	Unknown	1 (20.0)	1 (7.1)	1 (5.3)
All negative, *n* = 392	Any cause	30 (7.7)	80 (20.4)	91 (23.2)
	HCC-related	15 (50.0)	41 (51.3)	47 (51.6)
	Non-HCC-related	12 (40.0)	30 (37.5)	35 (38.5)
	Unknown	3 (10.0)	9 (11.3)	9 (9.9)

HCC, hepatocellular carcinoma; HCV, hepatitis C virus; HBV, hepatitis B virus.

## Data Availability

The data presented in this study are available in this article.

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
