# Peer review of "Causes of Death among Patients with Hepatocellular Carcinoma According to Chronic Liver Disease Etiology"

_cancers, 2023, doi:10.3390/cancers15061687_

Round 1

Reviewer 1 Report

In this manuscript, Authors investigated the causes of death in HCC patients, distinguishing HCC mortality from non-HCC mortality. Authors state that "although most patients with HCC die due to HCC-related causes, non–HCC related mortality represents a competing event in certain patient subgroups".

Some major concerns arise after evaluating the study:

- The aims of the study are not clearly reported. In the Introduction section, Authors state that the aim of the study was "to clarify the proportion of non–HCC-related deaths that occur among female patients with NAFLD-related HCC compared with other chronic liver disease (CLD) etiologies". However, the study did not address this specific aspect.

- Etiology of liver disease is not properly defined. Indeed, Authors defined the viral etiology just on the basis of anti-HCV antibodies or of HBsAg positivity, without giving any information about possible antiviral therapies that might have caused suppression of HBV replication or HCV clearance

- On which bases Authors used the described hierarchy?

- There are no data on possible metabolic etiology of liver disease, thus NAFLD/MAFLD etiology is not considered in this study

- Causes of death derived from death certificates, which are not enough accurate for the study purposes

- How did the Authors differentiate deaths due to decompensation of cirrhosis from those due to progression of HCC?

Author Response

In this manuscript, Authors investigated the causes of death in HCC patients, distinguishing HCC mortality from non-HCC mortality. Authors state that "although most patients with HCC die due to HCC-related causes, non–HCC related mortality represents a competing event in certain patient subgroups".

Some major concerns arise after evaluating the study:

- The aims of the study are not clearly reported. In the Introduction section, Authors state that the aim of the study was "to clarify the proportion of non–HCC-related deaths that occur among female patients with NAFLD-related HCC compared with other chronic liver disease (CLD) etiologies". However, the study did not address this specific aspect.

Response: Thank you so much for your comments. We have revised as follows: Furthermore, the proportion of non–HCC-related deaths that occur among patients with NAFLD-related HCC compared with other chronic liver disease (CLD) etiologies remains unclear. We aim to clarify this issue. Please see line 52-54.

- Etiology of liver disease is not properly defined. Indeed, Authors defined the viral etiology just on the basis of anti-HCV antibodies or of HBsAg positivity, without giving any information about possible antiviral therapies that might have caused suppression of HBV replication or HCV clearance

Response: We have acknowledged that the study has several limitations, including the following: Etiology of liver disease is not properly defined. We defined the viral etiology just on the basis of anti-HCV antibodies or of HBsAg positivity, without giving information about antiviral therapies that might have caused suppression of HBV replication or HCV clearance. Please see line 304-307.

- On which bases Authors used the described hierarchy?

Response: In line with the study conducted by Gawrieh et al., for patients with multiple etiologies, classification was performed using the following hierarchy: HCV > HBV > alcohol-related causes > all negative [9]. Please see line 75-77.

- There are no data on possible metabolic etiology of liver disease, thus NAFLD/MAFLD etiology is not considered in this study

Response: We have acknowledged that the study has several limitations, including the following: Furthermore, we did not have data on hepatic steatosis and metabolic risk factors (e.g., hypertension, dyslipidemia, central obesity, and hyperglycemia) [4]. Therefore, we could not define NAFLD in the present study. Please see line 300-302.

- Causes of death derived from death certificates, which are not enough accurate for the study purposes

Response: We have acknowledged that the study has several limitations, including the following: Causes of death derived from death certificates, which are not enough accurate for the study purposes. Please see line 307 and 308.

- How did the Authors differentiate deaths due to decompensation of cirrhosis from those due to progression of HCC?

Response: Since most patients with HCC who undergo treatment will develop recurrent tumors, decompensation of cirrhosis develops after multiple liver-directed therapies as well as due to tumor progression. In such a case, the cause of death indicated in the death certificate would be HCC rather than decompensation of cirrhosis. In our experience, however, few patients with HCC who have undergone treatment would die of decompensation of cirrhosis without tumor recurrence.

Reviewer 2 Report

The authors also provide an analysis and discussion of the all negative HCC group, which is well summarized. I have no further comments. I would recommend it for acceptance.

Author Response

The authors also provide an analysis and discussion of the all negative HCC group, which is well summarized. I have no further comments. I would recommend it for acceptance.

Response: Thank you so much for your comments.

Reviewer 3 Report

The study by Xi-Hao Yen and colleagues, by analysing a large cohort of almost 4000 patients, aims to answer an easy but crucial question about hepatocellular carcinoma (HCC), whether cancer is the main cause of death in HCC-patients or not. The article is well organized and quick to understand. Figures are clear and self-explaining. Results highlight how it is important not to underestimate comorbidities and potential non-HCC-related causes of death in specific conditions, such as patients with alcohol use disorders or older patients after undergoing curative treatments for HCC. The study has some limitations, notably the lack of definition of the different non-HCC-related causes of death and the presence of the inhomogeneous group of “all-negative patients” including NAFLD and rare hepatic diseases, but they are discussed in the final section of the manuscript.

Author Response

 Thank you so much for your comments.

Round 2

Reviewer 1 Report

Authors acknowledged many of the limitations reported in the previous review.